miR-344d-3p regulates osteogenic and adipogenic differentiation of mouse mandibular bone marrow mesenchymal stem cells

Cao Wei 1
Yang Xiaohong 854936065@qq.com 1
Hu Xiao Hua 2
Li Jun 3
Tian Jia 1
OuYang RenJun 1
Lin Xue 2
1 Department of Prosthodontics, Affiliated Stomatological Hospital of Zunyi Medical University, Zunyi Medical University , Zunyi , China
2 Department of Oral and Maxillofacial Surgery, Affiliated Stomatological Hospital of Zunyi Medical University, Zunyi Medical University , Zunyi , China
3 Department of Dental Implant, Affiliated Stomatological Hospital of Zunyi Medical University, Zunyi Medical University, Zunyi, Guizhou, China , Zunyi , China
Uversky Vladimir
Electronic publication date: 2023 Feb 14
Publication date: 2023
Volume: 11
Electronic Location ID: e14838
Received 2022 Jul 11; Accepted 2023 Jan 10
Copyright: ©2023 Cao et al.
Copyright year: 2023
Copyright holder: Cao et al.
License: This is an open access article distributed under the terms of the Creative Commons Attribution License, which permits unrestricted use, distribution, reproduction and adaptation in any medium and for any purpose provided that it is properly attributed. For attribution, the original author(s), title, publication source (PeerJ) and either DOI or URL of the article must be cited.
License URL: https://creativecommons.org/licenses/by/4.0/

Keywords: Osteoporosis, Mandibular bone marrow mesenchymal stem cells, miR-344d-3p, Dnmt3a, Osteogenic differentiation, Adipogenic differentiation

Funding: Natural Science Foundation of China 81660180 82060199 2022 Science and Technology Joint Fund Project of Zunyi Bureau of Science and Big Data and Zunyi Medical University (HZ (2022) 424) This work was supported by the Natural Science Foundation of China (No. 81660180, 82060199) and the 2022 Science and Technology Joint Fund Project of Zunyi Bureau of Science and Big Data and Zunyi Medical University (HZ (2022) 424). The funders had no role in study design, data collection and analysis, decision to publish, or preparation of the manuscript.

==============================
Postmenopausal osteoporosis (POP) is a chronic disease of bone metabolism that occurs in middle-aged and elderly women. POP can cause abnormalities of the skeletal system in the whole body, and the jaw bone is also impacted, affecting the function of the oral and maxillofacial regions. Mandibular bone marrow mesenchymal stem cells (MBMSCs) play an important role in mandibular bone metabolism, and abnormal differentiation of MBMSCs can affect the metabolic balance between new and old bone. MicroRNAs (miRNAs) can induce the differentiation of MBMSCs. In this study, the changes in biological characteristics of mandible and MBMSCs in the bone microenvironment of postmenopausal osteoporosis were firstly analyzed, and then the key miRNAs screened from miRNAs gene chips were sorted out for verification and functional exploration. It was found that miR-344d-3p promoted the osteogenic differentiation of MC3T3-E1 and MBMSCs. It inhibited the adipogenic differentiation of 3T3-L1 and MBMSCs. In addition, Dnmt3a may be the target gene of miR-344d-3p. In conclusion, this study found new biological indicators related to bone metabolism, which are of great significance in the field of bone reconstruction.

Introduction

The jaw bone is an important part of the whole skeletal system, which plays the role of supporting force, providing attachment and performing physiological functions of the cranium and face. Due to active remodeling physiological characteristics, jaw bone metabolic activities are also affected in the occurrence of systemic osteoporosis. result in bone mass loss and bone structure damage occur (Thanakun et al., 2019). Based on the fact that the main organ of estrogen production is ovary, and estrogen plays a role in promoting calcium absorption and bone synthesis in bone metabolism, many animal models of postmenopausal osteoporosis have been established, namely, inhibiting estrogen secretion by removing animal ovaries and affecting the balance of bone metabolism (Zhao et al., 2018). Studies have found that after bilateral ovariectomy in rats, the serum estrogen content decreases, and the content of calcium and alkaline phosphatase related to bone metabolism also decreases. The corresponding signs of bone loss can be observed through pathological sections (Wang et al., 2019). De Castro et al. (2020) conducted 3D reconstruction analysis of mandible of postmenopausal and non-menopausal women, and found that BMD, cortical thickness and other indicators of postmenopausal women decreased significantly. The result was consistent with clinical signs of postmenopausal osteoporosis (POP), and suggesting that the decrease of estrogen level may also cause abnormal mandibular bone metabolism to a certain extent. At present, the treatment of jaw osteoporosis mainly follows the treatment of systemic osteoporosis (OP), including non-drug treatment, such as good nutrition and regular physical activity; and drug therapy, such as estrogen agonist/antagonist, double phosphate and hardening protein inhibitors. But these measures are mainly focused on to prevent the happening of the OP. And medication also has some side effects, increased difficulties in diagnosis and treatment related to stomatology (Eastell et al., 2019; Kanis et al., 2019). Therefore, it is urgent to find a more advanced treatment method for the jaw after OP.

As the main source of proliferation and differentiation in bone tissue cells, bone marrow mesenchymal stem cells (BMSCs) play an important role in bone marrow due to their unique inducibility. It can specifically induce differentiation into osteocytes, adipocytes, chondrocytes and myogenic cells, etc., which is important in maintaining the body bone mass of organisms (Fu et al., 2019). Studies have shown that BMSCs tend to differentiate into adipocytes at the beginning, while the differentiation into osteoblasts decreases, resulting in lower BMD and OP (Qadir et al., 2020). In addition, BMSCs can be used as an ideal source of seed cells due to their ability to self-renew and differentiate into various cell types. BMSCs have been proved to be able to treat many diseases in many experiments (Guo et al., 2020; Jiang et al., 2021). In this study, we successfully established a mouse model of postmenopausal osteoporosis, and explored the changes in the biological characteristics of BMSCs derived from jaw bone. It was found that after POP, the osteogenic differentiation ability of MBMSCs was weakened, while the adipogenic differentiation ability was enhanced, which may be the main cause of the loss of jaw bone in mice.

It has been reported that more than 2,500 mature miRNAs have been found in the human genome, of which more than 60% have specific protein compilation targets. It regulates important physiological processes of organisms (Akhtar et al., 2016; Kozomara, Birgaoanu & Griffiths-Jones, 2019). BMSCs are also ideal cell carriers for miRNAs gene modification. MiR-21 (Yang et al., 2019a), miR-127 (Zhang et al., 2021) and miR-542-3p (Zhang et al., 2018) promote osteogenesis, while miR-31a-5p (Xu et al., 2018), miR-144-3p (Qiu et al., 2022) and miR-378 (Feng et al., 2020) promote lipid formation. The discovery of related target genes also elucidated the downstream pathway mechanism of miRNAs action and provided reference for us to understand the regulatory mechanism of miRNAs. Previous studies of our group found that miR-705 in mouse BMSCs could target Runx2 osteogenic genes by targeting Hoxa10 and Foxo1 and miR-3077-5p (Liao et al., 2016; Liao et al., 2013) to change the osteogenic differentiation function of BMSCs. On the contrary, the application of miRNAs related reagents in animal models of bone defects can achieve the results of bone regeneration and reconstruction. All of these discoveries suggest that the occurrence and development of OP can be controlled by changing miRNAs expression. It is worth that studies on miRNAs related to the pathogenesis of OP mainly focus on the long bone tissues of animals, while studies on the source of jaw bones are very few. However, skeletal cells in different parts of the body have different responses to biological signals (Berendsen & Olsen, 2015; Liu et al., 2020). Therefore, miRNAs in MBMSCs under POP state were sequenced by gene chip. After analysis and screening verification, miR-344d-3p was found to have the ability to regulate MC3T3-E1, 3T3-L1 cell lines and osteogenic and adipogenic differentiation of MBMSCs. And Dnmt3a may be the target gene of miR-344d-3p. This study provides important reference for future studies on the pathogenesis, diagnostic criteria and stem cell therapy of jaw OP, and is also of great significance in the field of oral prosthodontic-related bone reconstruction.

Materials & Methods

Experimental animals

Sixty female C57BL/6 mice aged 6 weeks (16 to 18g in weight), purchased from Chongqing Tengxin Biotechnology Co., LTD., License Number: SCXK (Beijing) 2019-0010, were randomly divided into Ovariectomies (OVX) group and sham-operated (Sham) group, with 30 mice in each group. The Affiliated Stomatological Hospital of Zunyi Medical University provided full approval for this research (No.: Lun Shen (2020) 2-473 and Lun Shen(2016)2-188). After separate cages, the animals were raised in SPF animal room with temperature of 26 °C and humidity of 50–65%. The animals could drink and eat freely. Mice were anesthetized with 1% pentobarbital sodium solution at 50mg/kg. Surgical methods of mice in OVX group: the middle and lower 1/3 of the back was skinned after anesthesia, and longitudinal incisions were made along the midline of the middle and lower short back of mice after iodophor disinfection. The skin, muscle and peritoneum were dissected in layers, the ovaries were found and removed, the remaining tissues were reset, and the muscle layer and skin layer were sutured contraptively. The procedure of SHAM group was the same as OVX group, but the ovaries were not removed. Mice were fed for 12 weeks for subsequent experiments.

Cell culture

Mice in the OVX group and the SHAM group aged 18 weeks after operation were sacrificed by cervical dislocation method, soaked in 75% alcohol, and then removed and placed in ultra-clean table about 5 min later. The mandible of mice was removed as soon as possible, and the attached soft tissues were removed. The mice were placed in α-MEM complete medium containing 20% fetal bovine serum, and the bone marrow cavity was rinsed and blown with a syringe. The remaining bone tissue of the mandible was cut and crushed until it became slag. The rinsed bone marrow together with the cut bone residue were placed into a 25 cm2 plastic culture flask and cultured at 37 °C in an incubator containing 5% CO2. After the cell growth and fusion reached 80%, it was digested with trypsin for passage. These MBMSCs were used for cell differentiation identification and miRNAs sequencing analysis. Blasts, Mouse embryo osteoblast precursor cells (MC3T3-E1), Mouse embryo fibroblasts, 3T3-L1 and 293T cell lines (Zhongqiao Xinzhou Co., Ltd, Shanghai, China) were cultured in a 25 cm2 plastic flask at 37 °C in an incubator with 5% CO2 concentration. MC3T3-E1 was cultured in 10% fetal bovine serum α-MEM complete medium. 3T3-E1 was cultured with DMEM complete medium containing 10% neonate calf serum, and 293T cell line was cultured with α-MEM complete medium containing 5% fetal calf serum. After cell growth and fusion reached 80%, trypsin was used for digestion and passage.

Cell transfection

Lipofectamine™ RNAiMAX (HanBio Co., Shanghai, China) transfected miRNAs on 24-well plates at a concentration of 40 nM (the procedure for transfection of miRNAs containing fluorescence is the same except that it needs to be carried out under dark conditions). 24 h before transfection, 2 × 105 cells were inoculated in medium without antibiotics containing 400 µL per well for transfection when the degree of cell confluency was 50%. Appropriate amounts of miRNAs mimics, inhibitors and blank controls were absorbed and added into EP tubes containing non-double resistant medium, respectively, and then gently blown and mixed. Lipofectamine™ RNAiMAX was absorbed into an EP tube, gently blown and mixed, and stood for 10 min at room temperature. The transfection complex was added to the 24-well cell plate to supplement the medium without double antibiotics, so that the final concentration was 40 nM, and the cell plate was evenly mixed by shaking the cell plate before and after. Pure medium was used for transfection without additional antibiotics. 4 to 6 h after transfection, the medium with serum containing double antibody was replaced. 48–72 h after transfection, it can be used for subsequent experiments.

Induced cell differentiation

When the cell growth and fusion reached 80%, the induced cells differentiated into osteogenic and adipogenic directions, respectively. The osteogenic induction solution was as follows: 1% penicillomycin +0.1 µM dexamethasone +10 mM β-sodium glycerophosphate +0.2 mM vitamin C. Lipid induction solution was configured as follows: 1% penicillomycin +1 µM dexamethasone +0.5 mM IBMX + 2 mM insulin +200 µM indomethacin.

Alizarin red staining and quantification

After osteogenic induction for 21 days, the cells were fixed with 4% paraformaldehyde for 0.5 h, washed with PBS for 3 times, and then added appropriate alizarin red dye (Solarbio, China), and incubated for 5 min. After the dye was washed with PBS, the cells were observed under a microscope and photographed. 500 µl 10% cetylpyridinium chloride was added to the stained cells, and the absorbance value was measured at 570 nm after standing at room temperature for 15 min.

Oil red O staining and quantification

After lipogenic induction for 10 days, the cells were fixed with 4% paraformaldehyde for 0.5 h, rinsed with PBS for 3 times, followed by adding 60% isopropyl alcohol for 10 min, immersed with oil red O dye for 15 min (Solarbio, Beijing, China), rinsed with PBS for 3 times, adding hematoxylin dye for 30 s, rinsed with PBS for 3 times, Oil red O Buffer was added and stood for 1 min. The dye was washed with PBS and observed under a microscope and photographed. Isopropyl alcohol was added to the stained cells, and the absorbance value was measured at 520 nm wavelength after standing at room temperature for 10 min.

qRT-PCR

According to the TRIzol specification, RNA was extracted from the sample (Yang et al., 2019b), and the system was configured with a reverse transcription kit (Bioengineering, China) to complete the reverse transcription reaction and obtain cDNA. The extracted RNA was subjected to agarose gel electrophoresis experiment, and then the electrophoresis samples were observed on the ultraviolet transmission tester. The bands of 28S and 18S samples were bright, clear and sharp, without trailing phenomenon, and the ratio was close to or more than 2:1. RNA with OD 260/280 > 1.8 and OD 260/230 > 1.5 can be used in subsequent experiments. Total RNA in RT reaction solution was 2 µg, and cDNA in PCR reaction solution was 2  µl. qRT-PCR amplification was performed using U6 and β-actin as internal reference for miRNAs and mRNA respectively (Biobiotics, China). Using tailing reaction to detect the expression of micrornas.

Western blot

Cells were lysed (RIPA: phosphatase inhibitor: PMSF = 100:1): Protein quantification and sample preparation were performed by BCA method (Solarbio, Shanghai, China). The protein loading volume was 20 µg. Gel preparation was performed by SDS-PAGE gel rapid preparation kit (Thermo Fisher, Waltham, MA, USA). After sample loading, electrophoresis, membrane transfer, sealing and primary antibody incubation were performed by steps (Abcam, Waltham, MA, USA). On the second day, second antibody incubation and exposure were completed (Millipore Corporation, China). After the bands were exposed, frame selection and gray value detection of swimming lanes were carried out by Image Lab software, and the protein data of target genes (such as RUNX2 and PPAR-γ) were compared with their internal reference, and then multiples of these data were compared with the data of control respectively.

Data analysis

SPSS 18.0 (SPSS, Inc., Chicago, IL, USA) and GraphPad Prism 9.0 (GraphPad Software, San Diego, CA, USA) was used for statistical analysis of data, and independent sample T test was used for comparison between the two groups, and one-way ANOVA was used for comparison between the three groups (nonparametric test and two-tailed test). All experiments were repeated at least three times, and representative experiments were shown. Differences were considered significant at P < 0.05.

Results

MBMSCs were imbalanced in osteogenic and adipogenic differentiation after POP

Alizarin red staining results of MBMSCs in the SHAM group and the OVX group after osteogenic induction showed that the number and volume of calcified nodules of MBMSCs in the SHAM group were larger than those in the OVX group (Fig. 1A). Quantitative results showed that the mineralization content of MBMSCs in the SHAM group was higher (P < 0.001) (Fig. 1B). qRT-PCR and WB results showed that the expression level of osteogenic related gene RUNX2 in MBMSCs in SHAM group was higher than that in OVX group (P < 0.001) (Figs. 1C and 1D). After adipogenic differentiation of MBMSCs in the SHAM group and OVX group, oil-red O staining results showed that the lipid droplets of MBMSCs in the OVX group were larger and more numerous than those in the SHAM group (Fig. 1E), and oil-red O quantitative results also showed that the lipid droplets in the OVX group were more abundant (P < 0.001) (Fig. 1F). qRT-PCR and WB results showed that the expression level of lippogenic differentiation related gene PPAR-γ in MBMSCs in OVX group was higher than that in SHAM group (P = 0.02) (Figs. 1G and 1H).

Figure 1 Analysis of osteogenic and adipogenic differentiation characteristics of MBMSCs in OVX group and SHAM group.

(A, B) Observation and quantitative analysis of Alizarin red staining of MBMSCs after osteogenic induction for 21 days. (C, D) mRNA and protein expression of osteogenic related gene RUNX2 of MBMSCs after osteogenic induction for 14 days. (E, F) Observation and quantitative analysis of oil red O staining of MBMSCs after 10 days of adipogenic induction. (G, H) mRNA and protein expression of lipid-related gene PPAR- γ in MBMSCs after 10 days of adipogenesis induction. SHAM, SHAM surgery; OVX, Ovariectomy; The scale bars in micrographs represent 100 µm and 50 µm from left to right; n = 3; * P < 0.05; ** P < 0.01.

MBMSCs showed low expression of miR-344d-3p after POP

Microarray sequence analysis was performed on MBMSCs of SHAM group and OVX group to screen out more than 20 miRNAs with highly differentially expressed genes. Significance analysis of microarrays (SAM) was used for differentially expressed genes. MiRNAs whose false discovery rate (FDR) is controlled within 5% and the difference multiple is not less than 4 times are screened out, and these miRNAs are arranged from high to low according to the difference multiple. Through literature review and analysis, miR-344d-3p with higher expression (most down-regulated in OVX group) was not reported (Figs. 2A and 2B).

Figure 2 Screening and verification of gene chip.

(A) Heat maps of miRNAs differential expression in MBMSCs of OVX group and SHAM group. (B) The statistical graph of differentially expressed miRNAs arranged by the size of difference multiples, among which miR-344d-3p (the most down-regulated in OVX group) had the largest difference multiples. (C) miR-344d-3p was down-regulated in MBMSCs of mice in OVX group. (D) miR-344d-3p was expressed endogenous in all tissues and differentially in adipose tissues. n = 3; ** P < 0.01.

qRT-PCR was used to detect the endogenous expression level of miR-344d-3p in MBMSCs in the SHAM group and OVX group, and it was found that miR-344d-3p was stably down-regulated in the OVX group, consistent with the chip results (P < 0.001) (Fig. 2C). Then, qRT-PCR was used to detect the expression levels of miR-344d-3p in mouse heart tissue, liver tissue, lung tissue, kidney tissue, pancreas tissue, adipose tissue and skeletal muscle, the values of OVX group were compared with those of SHAM group. The results showed that the expression levels of miR-344d-3p were significantly different in adipose tissue and low in other tissues, suggesting that miR-344d-3p may have adipose-specificity (Fig. 2D). Therefore, miR-344d-3p was selected as the target gene of this study.

miR-344d-3p has osteogenic induction to cells

Control, mimics and inhibitor with fluorescence of miR-344d-3p were transfected into MC3T3-E1 cell lines, respectively. After 48 h observation under inverted fluorescence microscope, green stains could be found in the cytoplasm (Fig. 3A). qRT-PCR results showed that in MC3T3-E1 cell line, the expression of miR-344d-3p in mimics group increased by about 30 times compared with control group (P < 0.001), and decreased by about 4 times in inhibitor group (P < 0.001) (Fig. 3B).

Figure 3 Analysis of the osteogenic induction of miR-344d-3p to MC3T3-E1 and MBMSCs.

(A) Fluorescence distribution under microscope after transfection of miR-344d-3p chemical complex into MC3T3-E1 cell line respectively. (B) qRT-PCR was used to detect the expression level of miR-344d-3p after chemical complex transfection with MC3T3-E1. (C, D) Alizarin red staining and quantitative analysis of the induction of miR-344d-3p to MC3T3-E1 osteogenesis. (E) Protein expression of osteogenic RUNX2 gene in MC3T3-E1. (F) Protein expression level of RUNX2 in MBMSCs. The scale bars in micrographs represent 100 µm and 50 µm from left to right; n = 3; ** P < 0.01.

After the miR-344d-3p chemical synthesis was transfected into MC3T3-E1 cell line, alizarine red staining results showed that the number of calcified nodules formed in the miR-344d-3p mimics group was the largest with large and deep volume, followed by the control group and the inhibitor group (Fig. 3C). Alizarine red quantitative analysis showed that the number of calcified nodules in miR-344d-3p mimics group was higher than that in control group (P < 0.01), and the number of calcified nodules in miR-344d-3p inhibitor group was lower than that in control group (P < 0.001) (Fig. 3D). WB results showed that the miR-344d-3p mimics group had the highest protein expression of RUNX2 (about 1.27 times higher than the control group), followed by the control group and the inhibitor group (0.18 times higher than the control group) (Fig. 3E). After transfection of the miR-344d-3p chemical complex into MBMSCs, WB results showed that the miR-344d-3p mimics group had the highest protein expression of RUNX2 (about 1.45 times higher than the control group), followed by the control group. The inhibitor group was 0.85 times the control group again (Fig. 3F). These results all suggest that miR-344d-3p positively regulates the osteogenic differentiation ability of MC3T3-E1 and MBMSCs.

Knockdown miR-344d-3p has adipogenic induction to cells

Control, mimics and inhibitor with fluorescence of miR-344d-3p were transfected into 3T3-L1 cell lines respectively, and observed under inverted fluorescence microscope 48 h later, green stains could be found in the cytoplasm (Fig. 4A). qRT-PCR results showed that in 3T3-L1 cell line, the expression of miR-344d-3p increased about 17 times in mimics group compared with control group (P < 0.001), and decreased about 2 times in inhibitor group (P = 0.01) (Fig. 4B).

Figure 4 Analysis of the adipogenic induction of miR-344d-3p to 3T3-E1 and MBMSCs.

(A) Fluorescence distribution under microscope after transfection of miR-344d-3p chemical complex into 3T3-L1 cell line respectively. (B) qRT-PCR was used to detect the expression level of miR-344d-3p after chemical complex transfection with 3T3-L1. (C, D) Oil-red O-red staining and quantitative analysis of miR-344d-3p’s lipid induction to 3T3-L1. (E) Protein expression of lipid-related gene PPAR- γ in 3T3-L1. (F) Protein expression level of PPAR- γ in MBMSCs. The scale bars in micrographs represent 100 µm and 50 µm from left to right; n = 3; * P < 0.05; ** P < 0.01.

After transfection of Figure chemical synthesis into 3T3-L1 cell line, oil red O staining results showed that the number of lipid droplets formed by miR-344d-3p inhibitor group was the most large and deep, followed by control group and mimics group (Fig. 4C). Quantitative analysis of oil-red O showed that the inhibitor group miR-344d-3p had higher lipid droplets than control group (P = 0.02), and the miR-344d-3p mimics group had lower lipid droplets than control group (P = 0.02) (Fig. 4D). WB results showed that PPAR-γ protein expression in miR-344d-3p inhibitor group was the highest (2.2 times higher than control group), followed by control group and mimics group (0.76 times higher than control group) (Fig. 4E). When the miR-344d-3p chemical complex was transfected into MBMSCs, the protein expression of PPAR-γ in miR-344d-3p inhibitor group was the highest (2.43 times higher than control group), followed by control group. The mimics group was 0.66 times higher than the control group (Fig. 4F). These results suggest that miR-344d-3p can negatively regulate the adipogenic differentiation ability of 3T3-L1 and MBMSCs.

Knockdown of miR-344d-3p increases the expression of Dnmt3a genes

Based on the analysis of biological information on target gene prediction websites (miRBase, miRand, miRWalk and TargetScan), it is predicted that miR-344d-3p may bind to the 3′-UTR region of more than 500 genes. Based on its function judgment, it is inferred that the target gene should have corresponding inverse function. Through literature screening and prediction of binding sites, it was found that there was a binding site between “TTATAT” sequence and miR-344d-3p in the 3′-UTR region of Dnmt3a (Fig. 5A). After transfection of miR-344d-3p chemical complex into 293T cell lines, protein expression of Dnmt3a target gene was detected by WB, and Dnmt3a protein expression was significantly increased in miR-344d-3p inhibitor group, while mimics expression was down-regulated (Fig. 5B). The above results indicated that miR-344d-3p overexpression prevented Dnmt3a protein expression, while endogenous Dnmt3a protein level increased significantly after down-regulation of miR-344d-3p. This suggests that miR-344d-3p blocks the translation of Dnmt3a target gene protein, and Dnmt3a may be the target gene of miR-344d-3p.

Figure 5 Target gene prediction and preliminary validation of miR-344d-3p.

(A) Website analysis and manual site detection found that there was A binding site between “TTATAT” sequence and miR-344d-3p in the 3′-UTR region of Dnmt3a. (B) WB results in 293T cell line showed negative regulation between miR-344d-3p and Dnmt3a, Dnmt3a protein expression was significantly increased and mimics expression was down-regulated in miR-344d-3p inhibitor group.

Discussion

With the increasing aging trend of the population, osteoporosis has become a common chronic disease that endangers the health of the elderly. The occurrence and development of a series of complications also increase the disability and mortality of such diseases. In middle-aged and old women’s groups, the drop in estrogen levels, makes the whole body bone metabolism abnormity, jaw is no exception, the loss of bone mass and difficult reconstruction, dentition defect and tooth missing and so on the treatment of oral disease outcome is affected, thus to explore the jaw bone metabolism in these diseases, provide new options for oral treatment measures and a new direction.

First of all, jaw bone differs from long bone in embryonic origin, growth and development, clinical manifestations of some bone-related diseases and sensitivity to clinical medication. Some studies have shown that after the treatment of OP with bisphosphonates, with the extension of treatment time, phosphoric acid deposits in bone will gradually accumulate and increase, resulting in an increased probability of osteonecrosis of jaw and absorption of molar roots (Vermeer et al., 2017). These results suggest that there may be differences in bone metabolism between jawbone and long bone, and stem cells play an important role in bone metabolism balance, so the mechanism of jaw bone bone metabolism needs to be explored separately. This is consistent with the relevant research conclusions that systemic POP can cause osteoporosis in the jaw bone as well as in the long bones of the limbs (Venkatesh et al., 2020).

RUNX2, an important transcription factor in osteogenic differentiation, plays an important role in bone remodeling. RUNX2 was weakly expressed in BMSCs as early as the differentiation of BMSCs into osteoblasts and the expression of RUNX2 gradually increased. RUNX2 regulates the proliferation and differentiation of osteoblast progenitors by directly regulating genes in classic bone metabolic signaling pathways such as hedgehog and Wnt (Komori, 2019). PPAR-γ is a kind of transcription factor with fat desiccation, which is closely related to fat metabolism. Its high expression can directly promote the differentiation of BMSCs to the adipogenic direction, and it is a landmark adipogenic related gene (Yuan et al., 2016). In this study, the expression differences of these two genes were mainly detected during cell differentiation. It was found that PPAR-γ was significantly up-regulated in MBMSCs under POP condition, while RUNX2 was on the contrary. This indicates that after estrogen deficiency, cell differentiation in jaw may be regulated by these two transcription factors, leading to more adipogenic cells. The loss of the original balance between osteogenic and adipogenic differentiation, increased fat accumulation, decreased the ability of new bone formation, resulting in the corresponding pathological manifestations of bone loss.

In order to explore the causes of abnormal differentiation of MBMSCs, this study focused on miRNAs that can regulate the post-transcriptional level, and obtained the miRNAs chip results exclusively on MOP by high-throughput sequencing technology, which was indeed different from the miRNAs chip results on long bone studied by our previous research group. Previous studies in our group found that miR-3077-5p and miR -705 in miRNAs chips of long bones can regulate BMSCs differentiation and affect the occurrence and development of osteoporosis after high-throughput sequencing (Liao et al., 2013). In this study, microarray analysis showed that miR-344d-3p had the highest possibility of differential expression in the two groups of cells, suggesting that different miRNAs may exist in different tissues. However, the application of miR-705 with differences in long bone chips to jaw bone can also affect the differentiation function of MBMSCs (Yang et al., 2019b). Therefore, it can also be speculated that the same miRNAs can act on both long bone and jaw bone. If the same miRNAs can be intervened, the subsequent bone-related mechanisms can be simplified and precise treatment measures can be studied. In this study, the regulatory performance of miR-344d-3p was firstly explored. miR-344d-3p showed that it could promote osteogenic differentiation and inhibit lipid differentiation of MBMSCs, which was consistent with the results of gene chip, indicating that miR-344d-3p plays an important role in jaw bone metabolism. It is worth noting that more than 20 miRNAs were found to be different in the chip results, which still needs to be verified and analyzed in order to have a more comprehensive understanding of the function of miRNAs and screen out miRNAs with promising research prospects.

Many studies have found that miRNAs can regulate the biological characteristics of BMSCs through specific target genes. For example, miR-21 promotes the osteogenic ability of BMSCs by activating the signaling axis of P-Akt and HIF-1 α (Yang et al., 2019a). MiR-705 changes the osteogenic ability of BMSCs by targeting Hoxa10 and Foxo1 (Liao et al., 2016) and miR-3077-5p targets RUNX2. Studies have shown that miRNAs participate in multiple signaling pathways related to bone metabolism, and combine gene targets to complete post-transcriptional regulation. Dnmt3a, a regulatory factor originally identified in DNA methylase, has been shown to play a role in a variety of diseases, including inflammation, immunity and tumor (Abplanalp et al., 2021; Brunetti, Gundry & Goodell, 2017). In recent years, it has been found that the expression of Dnmt3a increased during adipogenic differentiation of adipogenic mesenchymal stem cells and BMSCs (Stachecka et al., 2020). MiR-29a/B/C can inhibit 3T3-L1 lipogenic differentiation by targeting Dnmt3a (Zhu et al., 2017). MiR-371 can up-regulate the expression of Dnmt3a, while miR-369-5p can down-regulate the expression of Dnmt3a and play a role in the differentiation of human BMSCs (Zhu et al., 2017). The above studies indicated that the regulatory elements of Dnmt3a could bind to osteogenic or lipogenic promoters. It can activate the corresponding transcription activators such as RUNX2, PPAR-γ, ALP, etc., leading to changes in the expression levels of the corresponding factors and bone metabolism. It is well known that bone resorption and renewal is a process that depends on the function and activity of precursor osteoblasts. This process is regulated by a variety of extracellular signaling molecules, among which Dnmt3a is a key transcription factor, acting together with other factors to regulate cell differentiation. Interestingly, Dnmt3a was found to promote PPAR-γ methylation in MC3T3-E1-14 cells, thereby reducing its expression and promoting osteogenic differentiation (Lu et al., 2022). This is different from the results of this study, that is, the protein content of Dnmt3a decreased after the increase of bone formation ability of cells was detected in this study. This study believes that the reason may be that, first of all, Dnmt3a is indeed one of the factors that promote cell methylation, but the study of the above scholars first detected the change of Dnmt3a, followed by the decrease of PPAR-γ, at which time Dnmt3a has played its role. However, in this study, the increase of Dnmt3a after the cells had completed lipogenic differentiation could be the result of cell self-regulation. In addition, there may be thousands of target gene binding sites for miR-344d-3p, which was only verified in the preliminary exploration stage in this study. Further experiments such as luciferase reporter gene experiment may be more revealing. All these studies confirm that Dnmt3a is the promoter of transcription of important genes in bone regeneration and activates transcription of their genes, plays a pivotal role in the adipogenic differentiation phenotype of cells, is an important transcription factor for bone-related cell fate and differentiation, and is closely related to bone metabolism and bone disorders related diseases. In this study, protein analysis detected that Dnmt3a expression increased after down-regulation of miR-344d-3p. In addition, through the analysis of binding sites, it was found that there were two binding sites between the “TTATAT” gene sequence and miR-344d-3p in the 3′-UTR region of Dnmt3a, which further suggested that Dnmt3a might be the downstream target gene of miR-344d-3p regulating the differentiation of MBMSCs.

With the further study of epigenetics, the mechanism of bone metabolism has been elucidated gradually from various aspects of the network. On the basis of improving the miRNAs gene pool regulating the differentiation of MBMSCs, how to form a relationship with other regulatory factors in the estrogen-deficient bone microenvironment, deeply understand the mechanism of bone metabolism, and achieve the clinical effect of recovering bone loss still requires further research.

Conclusions

In our study, we found that the osteogenic differentiation ability of MBMSCs in postmenopausal osteoporotic mice was decreased, but adipogenic differentiation ability was enhanced, caused osteoporotic bone loss in jaw bone. miR-344d-3p promoted the osteogenic differentiation of MC3T3-E1 and MBMSCs. It inhibited the adipogenic differentiation of 3T3-L1 and MBMSCs. In addition, Dnmt3a may be the target gene of miR-344d-3p.

Supplemental Information

Supplemental Information 1 Author Checklist: Full

Click here for additional data file.

Supplemental Information 2 β-actin protein expression of MBMSCs after osteogenic induction

From left to right in lane 3 and 4

Click here for additional data file.

Supplemental Information 3 RUNX2 protein expression of MBMSCs after osteogenic induction

From left to right in lane 3 and 4

Click here for additional data file.

Supplemental Information 4 PPAR- γ protein expression of MBMSCs after adipogenic induction

From left to right in lane 5 and 6

Click here for additional data file.

Supplemental Information 5 β-actin protein expression of MBMSCs after adipogenic induction

From left to right in lane 5 and 6

Click here for additional data file.

Supplemental Information 6 RUNX2 protein expression of MC3T3-E1 after osteogenic induction

From left to right in lane 1 to 3

Click here for additional data file.

Supplemental Information 7 β-actin protein expression of MC3T3-E1 after osteogenic induction

From left to right in lane 1 to 3

Click here for additional data file.

Supplemental Information 8 RUNX2 protein expression of MBMSC after osteogenic induction

From left to right in lane 4 to 6

Click here for additional data file.

Supplemental Information 9 β-actin protein expression of MBMSC after osteogenic induction

From left to right in lane 4 to 6

Click here for additional data file.

Supplemental Information 10 PPAR- γ protein expression of 3T3-L1 after adipogeni induction

From left to right in lane 1 to 3

Click here for additional data file.

Supplemental Information 11 β-actin protein expression of 3T3-L1 after adipogeni induction

From left to right in lane 1 to 3

Click here for additional data file.

Supplemental Information 12 PPAR- γ protein expression of MBMSCs after adipogeni induction

From left to right in lane 1 to 3

Click here for additional data file.

Supplemental Information 13 β-actin protein expression of MBMSCs after adipogeni induction

From left to right in lane 1 to 3

Click here for additional data file.

Supplemental Information 14 Dnmt3a protein expression of 293T cells

From left to right in lane 1 to 3

Click here for additional data file.

Supplemental Information 15 β-actin protein expression of 293T cells

From left to right in lane 1 to 3

Click here for additional data file.

Supplemental Information 16 miR-344d-3p sequence

miR-344d-3p sequence from riginal sequencing results

Click here for additional data file.

Supplemental Information 17 The sequencing results: all miRNA

Click here for additional data file.

Figure S1 All raw data from Figure 1

Raw data: Figure 1 A-H

Click here for additional data file.

Figure S2 All raw data from Figure 2

Raw data: Figure 2 A-D

Click here for additional data file.

Figure S3 All raw data from Figure 3

Raw data: Figure 3 A-F

Click here for additional data file.

Figure S4 All raw data from Figure 4

Raw data: Figure 4 A-F

Click here for additional data file.

Figure S5 All raw data from Figure 5

Raw data: Figure 5 A-B

Click here for additional data file.

Additional Information and Declarations

Competing Interests

Author Contributions

Animal Ethics

DNA Deposition

Data Availability

The authors declare there are no competing interests.

Wei Cao conceived and designed the experiments, performed the experiments, analyzed the data, prepared figures and/or tables, authored or reviewed drafts of the article, and approved the final draft.

Xiaohong Yang conceived and designed the experiments, analyzed the data, authored or reviewed drafts of the article, and approved the final draft.

Xiao Hua Hu analyzed the data, prepared figures and/or tables, and approved the final draft.

Jun Li analyzed the data, prepared figures and/or tables, authored or reviewed drafts of the article, and approved the final draft.

Jia Tian performed the experiments, prepared figures and/or tables, authored or reviewed drafts of the article, and approved the final draft.

RenJun OuYang performed the experiments, prepared figures and/or tables, authored or reviewed drafts of the article, and approved the final draft.

Xue Lin performed the experiments, prepared figures and/or tables, authored or reviewed drafts of the article, and approved the final draft.

The following information was supplied relating to ethical approvals (i.e., approving body and any reference numbers):

Affiliated Stomatological Hospital of Zunyi Medical University provided full approval for this research (No.: Lun Shen (2020) 2-473 and Lun Shen(2016)2-188).

The following information was supplied regarding the deposition of DNA sequences:

The data is available at GenBank: RUNX2 sequences, PRJNA233974; PPAR-γ sequences, PRJNA705462; β-actin, PRJNA98285; Internal U6, PRJNA381487.

The miR-344d-3p sequencing results are available in the Supplemental Files.

The following information was supplied regarding data availability:

The raw measurements are available in the Supplemental Files.

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
