# Peer review of "miR-344d-3p regulates osteogenic and adipogenic differentiation of mouse mandibular bone marrow mesenchymal stem cells"

_PeerJ, doi:10.7717/peerj.14838_

## Round 0.1 · original submission · Major Revisions

Please address in detail the experimental design issues raised by Reviewers 1 and 2, and the 'validity of the findings' issues by all 3 reviewers. In addition, please submit the entire images for figures 1 and 15. For figure 9, the heatmap is incompletely cropped- please consider reproducing the figure using publicly available data. For your resubmission, it is recommended that you address every comment from the reviewers line-by-line, and include the changes that you have made in the revised version.

In addition, the manuscript requires substantial changes with formatting and grammatical corrections-

1. The figures need to be combined into panels, depending on their relevance to the results. For example, figures 1-5 can be summarized into one panel.
2. Please add a schematic of miR-344d-3p in regulating Dnmt3a to explain the mechanism that you validate from your findings.
3. The axes in the figures need to be checked and labeled properly.
4. The reference style should be consistent throughout the manuscript.
5. For grammar and language corrections, please consider using a proofreading service.

Reviewer 1 ·

Basic reporting

This manuscript reported that osteogenic and adipogenic abilities of MBMSCs derived from OVX mice were reduced and increased, respectively, compared to the SHAM control. Further analysis found that miR334d-3p expression was down-regulated in MBMSCs of mice in the OVX group compared to the control. Using miRNA mimic and inhibitor, authors reported that overexpression or knockdown of miR334d-3p resulted in increased and decreased osteogenic and adipogenic abilities, respectively. They found overexpression or knockdown of miR334d-3p could affect Dnmt3a expression. The results are interesting and novel. However, some concerns need to be clarified before publication as follows.

Experimental design

The experimental design was acceptable.
New experiments to knockdown and overexpress Dnmt3a should be performed to determine if Dnmt3a can affect osteogenic and adipogenic differentiation of MBMSCs. Providing the results will greatly strengthen the manuscript and conclusion.

Validity of the findings

Some concerns are listed below for the authors to improve the manuscript.
(1) Provide some background information about OVX to induce POP.
(2) Lines 107 to 121: clarify the age of the mice used for cell isolation. It was mentioned that the remaining bone tissue was collected on lines 112-113; however, there was no description regarding using the bone tissue for any experiments.
(3) Lines 123 to 136: several places used “ non-double resistant medium” please explain what the medium was and why the medium was used.
(4) Line 147: Please clarify the percentage and volume of cetylpyridine Chloride added to the cells (each well). I believe it should be cetylpyridinium Chloride, not cetylpyridine Chloride. Correct?
(5) Lines 157-170: Please provide more information about the RT-PCR and Western blot: (a) how to ensure the RNA quality. (b) How much RNA and cDNA were used for RT and PCR, respectively? (c) please provide the Primer sequences information. (d) was stem-loop RT-PCR used to detect miRNA expression? (d) how much protein was used for western blot?
(6) Figures in the Results section were grouped and described (Figures 1-5); however, they were shown as individuals (Figures 1-26). Please group them as described in the Results section.
(7) For western blot figures (4, 8, 17, 18, 23, 24, and 26), please describe the calculation in the legend of how the numbers shown under the photos were obtained.
(8) Non-transfection controls should be provided to ensure there was no autofluorescence in Figures 13 and 19.
(9) Figures 1 and 15, in addition to the high-power images, please provide images to show the entire wells for a better perception of the results.
(10) miR344d-3p mimic and inhibitor transfect experiments were conducted with MBMSCs to assess its effect on osteogenic and adipogenic differentiation, but only expression of differentiation markers (Runx2, PPAR-ɣ) was reported. Why not provide the ARS ad ORO staining images of the experiments?
(11)In the discussion section, please discuss studies on cell differentiation reported in the literature regarding Dnmt3a.

Additional comments

Engish should be polished by professional editing service.

·

Basic reporting

The authors were able to convey their statements based on the data observed and report the findings. Here are some suggestions specifically for the introduction and reference parts:

1. The reference style should be consistent throughout the paper. It will be easier for readers to match the exact publication referring to at the reference page.
2. In the introduction section, the authors tend to combine several short sentences into one long sentence, making it hard for readers to digest and comprehend the points the authors try to deliver. I suggest rewrite the introduction and separate them into short sentences and add the proper reference right after each sentence.

Experimental design

The authors were able to design a consequential experiments based on the phenotypic differences observed in two groups, starting from testing osteogenesis and adipogenesis markers, performing a miRNA screening, validating the function of the target miRNA, proposing a downstream target and potential pathways it's involved in. Here are some feedbacks for the experimental design:

1. In Figure 1, the authors showed data 21, 14, and 10 days after osteogenic induction in different sub-figures and the reason for this time difference is not explained. Since the authors claim the both osteogenic and adipogenic characteristics are the direct effect of the treatment, it is critical to address the difference in such dynamic.

2. The mimics and inhibitor used in Figure 3 and 4 should be better explained, how to mimic and how to inhibit. Especially the mechanism of inhibition is critical to assess Figure 5 the regulation of Dnmt3a expression.

3. The author mentioned that miR-344d-3p is predicted to bind to about 500 genes. At least the top 10 hits should be provided and what are the other genes with the same binding motif "TTATAT" as Dnmt3a. It is important to show that Dnmt3a is the key regulator or there are other genes with higher binding. In addition, if that's the case, the authors should demonstrate another two targets other than Dnmt3a in order to claim the inverse regulation of miR-344d-3p to the expression of targets it binds to.

Validity of the findings

The authors carefully analyzed the data in this study. Most of the data required to assess the data is provided by the authors. However, there are some missing pieces and some control groups to add that could strengthen the authors claim. Here are the feedbacks:

1. For analyzing the osteogenic and adipogenic differentiation, I suggest to add a group of "no osteogenic induction" to properly evaluate the differential response of the two groups to such induction.

2. It would be more transparent to indicate the sample size for all the bar graphs as well as the individual value. Statistically, I suggest the author to specify exactly what at the type of t-test throughout the study (parametric vs nonparametric / one-tailed vs two-tailed). Many pieces of evidence are on the verge of being statistically significant and variance between groups appears to be different, making the type to t-test essential to assess this finding.

3. For Figure 2D, the authors expression of miR-344d-3p in different tissues, suggesting adipose has the highest abundance. To continue from Figure 2C in stating the important of adipose tissue in such response, I suggest to perform the same experiments in both SHAM and OVX group. It is important to see whether the decrease of miR-344d-3p is observed in all tissues or just adipose, and therefore rationalize the following focus on adipose tissue.

4. In Figure 3A, the authors mentioned that the green is the fluorescence tag of miR-344d-3p; however, from the green intensity seen in Figure 3A, it is not consistent with the quantification of miR-344d-3p (mimics group has lower green intensity in 3A but expression is 20-fold higher in 3B). I suggest to show the representative data with the same exposure duration and indicate the transfection efficiency among the samples. In addition, the authors mentioned the fluorescence was observed in both nucleus and cytoplasm. As the later part to the paper is claiming its function in RNA binding and regulation which occurs in the cytoplasm, Figure 3B can support the concept more by provide the difference in fractions of nucleus and cytoplasm.

Additional comments

The authors demonstrated the regulation of miR-344d-3p in response to osteogenic induction and suggested the potential downstream targets. This is a novel finding reporting the role of miR-344d-3p in gene regulation. However, the rationale about focusing on certain genes and pathways is unclear, which could be further strengthened by providing a comprehensive screen results and validating more genes to claim a general molecular function.

Reviewer 3 ·

Basic reporting

The manuscript by Cao et.al has efficiently utilized molecular biology techniques to uncover the function of mir-344d-3p in MBMSCs. The introduction and discussion sections are well-written in professional English. The figure section lacks proper organization and structure to reflect the references in the text, all figures need to be assembled into main figure with sub-panels. Figure legends are not supposed to appear in the main text, instead it should be either an independent section or accompanying the corresponding figures. The methods and results sections have a few typographical errors which should be re-checked. The authors need to include more introduction on the Dnmt3a gene, to give the readers a better idea about the physiological relevance of the mir-344d-3p and Dnmt3a pathway axis.

Experimental design

no comment

Validity of the findings

In Figure 5, the authors concluded in the section header that "Knockdown of miR-344-3p increases the expression of Dnmt3a genes in MBMSCs". However, the experiments were performed in 293T cell lines. The authors need to clarify this discrepancy.

Additional comments

I recommend that the author polish the grammar to improve the readability of the manuscript and meet the publishing standards.
One suggestion is to make a cartoon/schematic representation of the proposed model of interplay between miR-344d-3p and Dnmt3a in human breast cancer.
The authors need to make sure that the axises are properly labeled.

---

## Round 0.2 · Major Revisions

Please note that although reviewer #2 was satisfied by the revision, reviewer #1 requires conducting additional experiments and providing some explanations. Please address all remaining concerns of this reviewer and revise manuscript accordingly

Reviewer 1 ·

Basic reporting

The revision has addressed some of my concerns, but there are still some problems to be addressed before accepting it for publication. Please find my comments in the attachment.

Experimental design

I recommend a new experiment.

Validity of the findings

See attachment

Annotated reviews are not available for download in order to protect the identity of reviewers who chose to remain anonymous.

·

Basic reporting

The authors were able to address the questions raised by reviewers and enhance the clarity of the story.

Experimental design

no comment

Validity of the findings

no comment

---

## Round 0.3 · Minor Revisions

Please address the remaining concerns of the reviewer and amend the manuscript accordingly.

Reviewer 1 ·

Basic reporting

I think a better title would be: “miR-344d-3p regulates osteogenic and adipogenic differentiation of mouse mandibular bone marrow mesenchymal stem cells” or “Regulation of osteogenic and adipogenic differentiation of mouse mandibular bone marrow mesenchymal stem cells by miR-344d-3p”
I suggest the titles because the manuscript did not provide strong evidence to support that miR-344d-3p promoted osteogenic differentiation via inhibiting (targeting) Dnmt3a expression. Given that miR-344d-3p is predicted to target 814 genes by miRDB, and some studies reported DNMT3a promotes osteoblast differentiation (Lu et al., 2022), it is likely miR-344d-3p promotes osteogenic differentiation and inhibits adipogenic differentiation shown in this study by targeting other genes.

Experimental design

No further comment

Validity of the findings

No comment

---

## Round 0.4 · accepted · Accept

Thank you for addressing the remaining concerns of the reviewer. I am pleased to accept the revised version for publication now.